# Towards the Influence of Media on Suicidality: A Systematic Review of Netflix’s ‘Thirteen Reasons Why’

**DOI:** 10.3390/ijerph20075270

**Published:** 2023-03-27

**Authors:** Martí Guinovart, Jesús Cobo, Alexandre González-Rodríguez, Isabel Parra-Uribe, Diego Palao

**Affiliations:** 1Department of Mental Health, Hospital Universitari Parc Taulí, 1 Parc Taulí, 08208 Sabadell, Spain; 2Centro de Investigación Biomédica en Red de Salud Mental (CIBERSAM), Instituto de Salud Carlos III, 3-5 Calle Monforte de Lemos, Pabellón 11, Planta 0, 28029 Madrid, Spain; 3Department of Psychiatry and Forensic Medicine, Universitat Autònoma de Barcelona, Plaça Cívica, 08193 Bellaterra, Spain; 4Department of Mental Health, Hospital Universitari Mútua de Terrassa, 5 Plaça del Doctor Robert, 08221 Terrassa, Spain; 5Department of Psychiatry, University of Barcelona, 585 Gran Via de les Corts Catalanes, 08007 Barcelona, Spain

**Keywords:** ‘Thirteen reasons why’, Werther effect, suicide, media, self-harm

## Abstract

**Highlights:**

Fictionalized suicides have shown potential for suicide contagion.‘Thirteen reasons why’, released in 2017, has been reported to increase suicidality.Positive and negative outcomes regarding suicide have been associated with the show.Individual factors define vulnerable viewers that should be warned of the risk.

**Abstract:**

Online streaming series ‘Thirteen Reasons Why’ (13RW), released in March 2017, was criticized for its sensationalist portrayal of the main character’s suicide, leading some people to voice fears of a global contagion of self-harm behaviors. The current investigation provides a systematic review of original studies analyzing the role of 13RW as an influencing factor for suicide. Articles were identified through a systematic search of Medline, Web of Science, Scopus, PsycInfo, and a manual search of reference lists from inception until the 16 January 2023. Twenty-seven published articles were identified from an initial search of 496 studies. The positive effects of watching 13RW included a reduction in suicide stigma and a greater likelihood to discuss mental health concerns and seek for help. However, several studies reported negative outcomes, including significant increases in the rate of deaths by suicide in adolescents, the number of admissions for suicidal reasons, and the prevalence and severity of suicidal ideation and self-harm behaviors in vulnerable viewers. Still, due to methodological limitations, no causal relationship could be established. Preventive measures are required to alert of the risk and should be particularly addressed to susceptible subjects. Psychoeducational programs should be focused on this kind of phenomena in vulnerable populations.

## 1. Introduction

It was 31 March 2017 when Netflix, an online streaming platform, released the first season of ‘Thirteen Reasons Why’ (13RW). The thirteen-chapter series was based on the multi-award winning novel of the same name, a text which had been written by Jane Asher in 2007 and which sold more than 3 million copies [1,2].

The show depicts the suicide of Hannah Baker, a female adolescent who, prior to death, decides to record thirteen audiotapes blaming several people for having pushed her to the edge. The portrayal of the main character is presented in a tragic, often glamorized way, and emphasizes the influence of environmental factors on suicide decision making. The last chapter contained a highly explicit scene showing how she finally manages to die by wrist cutting in the bathtub. The show assumes several conflictive premises (see Table 1) and this raises concerns as to how the series’ approach to suicide might induce viewers to imitate this behavior.

Suicide contagion through media, a phenomenon known as the ‘Werther effect’, can be approached if we hypothesize that media depictions of suicide might induce vulnerable viewers to mimic this behavior, especially sensationalist descriptions including vivid details [3]. Several previous experiences have shown confirmative results on the field, reporting an increase in suicide rates after the tragic deaths of Marilyn Monroe [3,4] and Robin Williams [5,6], among other celebrities. These copycat gestures tend to present themselves in the first weeks after the index episode and are usually exhibited by people with the same age, gender and sociodemographic profile who, these similarities aside, often use the same method [7,8,9,10,11]. Depression and negative life events are also risk factors for the imitation of celebrity suicides publicized in the media [9,10].

Not only real-life suicides have been suggested to have contagious potential. In the case of fictional suicides, some authors have pointed out that vulnerable subjects, when exposed to films that depict suicide, experience a rise in suicidality that correlates with the level of identification with the main character [12,13,14,15]. One week after TV series ‘Casualty’ depicted the deliberate self-poisoning of a teenager, overdose presentations increased by 17% in the UK, with 20% of patients declaring that the program had influenced their decision [16]. Similar reports regarding TV-dramatized adolescent suicides have been described in the USA [17] and Germany [18]. As the targeted audiences for 13RW mainly comprise adolescents and young adults, it would be sensible to consider that viewers of such ages are the population at risk of mimicking self-harm gestures after watching the series. In the era of social networking sites, parasocial relationships (and their influence on suicide models) need to be taken into consideration, especially in younger populations [19,20,21].

On the other hand, it is also true that other experiences where a Werther effect would have been expected did not show such outcomes. The suicide of Kurt Cobain, for example, as opposed to the celebrity deaths mentioned above, was not followed by a significant increase in suicide mortality [22,23]. As positive results are socially alarming and, consequently, more likely to be accepted in press, a publication bias cannot be discounted. Debate on the matter was already ongoing in the late 1980s, with some studies finding no reliable association between television coverage of suicides and self-inflicted deaths in adults [24] and teenagers [25]. A more recent meta-analytic review also concluded that current data do not support the theory of suicide contagion by fictional media [26]. Such contradictions suggest that the phenomenon—in the case it exists—is far more complex and nuanced than it appears at a simple view and depends of multiple, individualized, hardly controllable factors.

To avoid imitative behaviors, the World Health Organization (WHO) developed a booklet for media professionals that proposed how to treat suicide cases properly. This full-access resource was created in 2008 and last updated in 2017 [27]. These guidelines advise against using sensationalist language and recommend not to provide explicit graphic support or specific details about the site/method used. As responsible reporting about suicide can contribute to self-harm prevention (the so-called ‘Papageno effect’ [28], in contrast to the Werther effect), providing accurate information about where to seek help and educating the public without spreading myths are strategies that should be promoted. The tone and esthetics of the series, however, diverge significantly from these postulates.

The series obtained such positive reviews from critics and audiences that, shortly after, a second season premiered to expand the story. The continuation of 13RW aired on the 18 March 2018 but, unlike the first season, alerting videos and warning cards were added at the beginning of each episode informing of the treated topics. An informative documentary, entitled ‘Thirteen Reasons Why: Beyond the Reasons’, was also released: a special program where the cast, screenwriters, producers and mental health professionals discussed scenes dealing with difficult issues. As forthcoming evidence seemed to support the increase in suicide risk following exposure to 13RW, and in the light of the rising social alarm, the suicide scene in the last episode of season one was finally edited in July 2019.

The story was extended into a third and a fourth season. However, these final chapters deviated significantly from the original story, had nothing to do with suicide and received generally negative reviews from both critics and audiences.

In any case, 13RW represents the first reported case of a dramatized suicide being broadcasted from an online streaming platform. Streaming services have represented a change in the pattern of content consumption as they offer the opportunity to binge-watch the episodes and to replay certain scenes over and over [29]. In consequence, it would be sensible to analyze to what degree suicide contagion transmitted through Netflix may have differed from previous instances with more traditional media (literature, press, television, or cinema) [30].

### Objectives of the Study

In this study, we aimed to provide a systematic review of scientific publications that have linked 13RW with any outcome related to suicidality, considering data obtained from (1) objective sources (such as suicide mortality rates, suicide attempts rates, or registers of information seeking behaviors for suicide in any context) and (2) subjective perceptions from 13RW viewers (such as self-reported suicidal ideation or attempts, behavioral changes attributed to the show, or suicide knowledge and stigma). 

Additionally, we aimed to determine if current evidence on 13RW supports the presence of a Werther (pro-suicidal) or a Papageno (anti-suicidal) effect.

Finally, we aimed to assess whether individuals with the same features as 13RW’s protagonist (adolescent females) have been influenced differently by the possible effects of the show when compared to other subjects.

## 2. Methods

This review followed the PRISMA (Preferred Reporting Items for Systematic Reviews and Meta-Analyses) guidelines [31].

### 2.1. Inclusion and Exclusion Criteria

‘Suicide’ was defined as the act of taking one’s own life voluntarily with the intent to die. ‘Suicidality’ is a broader concept that comprises a spectrum of symptoms, from suicidal ideation to actual suicide attempts, fatal or not [32,33].

To be included, articles had to meet the following eligibility criteria: (1) be original research articles; (2) be written in English, Spanish or Catalan; (3) be performed in humans; (4) consider ‘having watched any episode of any season of 13RW’ as the exposure of interest; (5) report any effect related to suicidality; and (6) express outcomes in quantitative terms.

Exclusion criteria were, as follows: (1) be articles with non-empirical designs (e.g., case reports, reviews, book chapters, opinion letters, commentaries, conference papers or meeting abstracts); (2) be written in other languages apart from English, Spanish or Catalan; (3) be performed in vitro or in animals; (4) focus on other exposures (e.g., having read the book on which 13RW is based, or watched any other show related to the issue, but not specifically 13RW); (5) analyze other effects in the field of mental health, but not specifically suicidality (e.g., depression, anxiety or resiliency); and (6) express results in qualitative terms.

The presence of a reference group was not required. No restrictions were considered in terms of age, gender, ethnicity, country or publication year.

### 2.2. Search Strategy

Four bibliographic databases (Medline, Web of Science, Scopus, and PsycInfo) were assessed up to the 16 January 2023. The search strategy used the terms: (“thirteen reasons why” OR “13 reasons why”) AND (suicide OR suicidal OR suicidality). The reference lists of the included studies and other relevant papers were manually searched for additional articles.

Two independent systematic searches were conducted. Working by title and abstract reading, two reviewers (blinded to each other’s decision) independently completed article selection. After all the studies had been screened, the researchers discussed the results article by article for inclusion and, in cases of opinion disparity, consensus between reviewers prevailed. Selected references were full-text read for a second screening before proceeding to data extraction.

### 2.3. Risk of Bias and Quality Assessment from Included Studies

The Appraisal tool for Cross-Sectional Studies (AXIS tool) is a 20-item critical appraisal instrument which was created to address study design and reporting quality as well as the risk of bias in cross-sectional studies. It was developed in 2016 and with the aim of incorporating it in systematic reviews, guidelines and clinical decision making [34,35].

In the present study, this tool was used to assess the quality and risk of bias for included articles. As in article selection, two reviewers scored all the included studies separately and, afterwards, they compared their results. In cases of disparity, each item of the scale was discussed by both reviewers until an agreement was reached.

### 2.4. Data Extraction and Data Synthesis from Included Studies

The following data from the selected studies were recorded in a pre-designed form: (1) study author and publication year; (2) dependent variable(s); (3) participants, recruitment/source and setting; (4) study design; and (5) main outcomes. Data extraction was conducted by two authors and, in cases of doubt, results were discussed with two other investigators and solved by consensus from all authors.

Preliminary searches showed that, according to their design, studies could be divided into two categories: (1) studies reporting outcomes based on *objective* measures, such as population registers, hospital attendance rates, or prevalence of use of certain resources; and (2) studies reporting outcomes based on *subjective* measures, which basically means self-reported questionnaires from viewers of the show. Objective measures offer more consistent evidence and are less prone to interpretation. Subjective measures, on the other hand, are highly dependent on the sample that has been chosen and should be addressed more carefully, considering the risk of bias. Accordingly, results were synthetized in two different charts, one for objective data and another for results obtained with self-reported tools.

The considerable between-study heterogeneity hampered the ability to perform a meta-analysis. Therefore, a narrative synthesis of the included articles is presented.

## 3. Results

The comprehensive electronic search strategy yielded 490 potentially relevant studies. Six further papers were added from the reference list hand-search (see Figure 1). After removing duplicates and screening for title, abstract, and full-text, 27 non-overlapping articles contributed to the final data synthesis.

### 3.1. Outcomes in Association with 13RW Based on Objective Measures (See Table 2)

Fourteen studies were identified and all of them had been performed in the USA or Canada.

**Table 2 ijerph-20-05270-t002:** Summary of the articles reporting outcomes in association with 13RW based on objective measures.

Article	Dependent Variable(s)	Sample	Design	Results
Ayers et al., 2017 [36]	Google searches including the term ‘suicide’	Search trends with the term ‘suicide’ (data extracted from Google Trends), USA	Ecological study comparing Google searches for suicide in the 19 days after the release of 13RW’s first season with expected values	- 1.5 million excess searches (+19%). - Examples: “how to commit suicide” (+26%), “commit suicide” (+18%), “how to kill yourself” (+9%), “suicide hotline number” (+21%), “suicide hotline” (+12%), “suicide prevention” (+23%) and “teen suicide” (+34%).
Bridge et al., 2020 [37]	Number of suicides (global)	180,655 suicides occurred in individuals aged 10 to 64 years in the period 2013–2017 (data extracted from the Centers for Disease Control and Prevention WONDER system), USA	Ecological study comparing suicide rates in the 9 months after the release of 13RW’s first season with expected values	- 195 excess suicides among 10- to 17-year-olds in the 9 months after the release, with higher impact in males. - 28.9% increase in the first month after the release. - No excess suicide mortality in other age groups.
Cooper et al., 2018 [38]	Hospital admissions for suicidal ideation or self-harm with intent to die (youths)	775 admission records from patients aged 4 to 18, hospitalized for suicidal ideation or self-harm with the intent to die from January 2012 to October 2017 in The Children’s Hospital at OU Medical Center, Oklahoma, USA	Ecological study comparing hospitalizations for suicidal reasons in the 7 months after the release of 13RW’s first season with expected values	- 0.024 increase in the log of suicide admissions per month in the overall period. - The model that best explained the data indicated an increase in the number of admissions following the show’s release.
Feuer and Havens, 2017 [39]	Emergency department admissions for suicidal ideation or suicide attempts (youths)	Data obtained from surveys addressed to 14 pediatric emergency services, USA	Survey administration after the release of 13RW’s first season	- 95% sites reported significant increases compared to same month previous year. - 40% sites reported copycat gestures or attempts in the 30 days after the release.
Le et al., 2022 [40]	Emergency department and hospital admissions for suicidal behavior (youths)	289,350 hospitalizations and 815,582 emergency department visits for suicidal behavior from patients aged 6 to 17 in the period 2016–18 (data extracted from the Nationwide Inpatient Sample and the Nationwide Emergency Department Sample of the Healthcare Cost and Utilization Project), USA	Ecological study comparing emergency department and hospital admissions for suicidal behavior in the 21 months after the release of 13RW’s first season with expected values	- Emergency department visits for suicidal reasons: increase in the first and/or the second month after the release for all 10- to 17-year-olds (males and females). - Hospital admissions for suicidal reasons: increase in the second month after the release for all females aged 10 to 17 (especially for Black females) and for those aged 6 to 9 who identified as “Other” races/ethnicities. No increase in males.
Niederkrotenthaler et al., 2019 [41]	Number of suicides (global)	Suicide data from the period 1999–2017 (data extracted from the Centers for Disease Control and Prevention WONDER system), USA	Ecological study comparing suicide rates in the 3 months after the release of 13RW’s first season with expected values	- 66 excess suicides among 10- to 19-year-old males (+12.4%) and 37 among females (+21.7%). - No excess suicide mortality in other age groups. - 26.9% increase in the hanging method, no increase in the firearm method.
Pérez-Martínez et al., 2020 [42]	Published comments on Twitter about 13RW related to suicide	151,570 tweets containing the words #13ReasonsWhy and/or #PorTreceRazones (data obtained with the platform Followthehashtag), worldwide	Retrospective analysis of the content of Twitter posts in the 7.5 months after the release of 13RW’s second season	- Highest publication frequency in the first month after the release. The USA (43.7%) and the UK (13.5%) ranked first in the subject of suicide. - 32% report that 13RW does not adequately address suicide and 52.3% that it contributes to suicide debate.
Plager et al., 2019 [43]	Clinical context where references to 13RW are presented	63 clinical notes from 31 patients, aged 11 to 17, attending Fairview Health Services, Minnesota (found by searching the University of Minnesota-affiliated Fairview Health Services clinical data repository), USA	Retrospective analysis of documented references to 13RW in pediatric patients from October 2015 to September 2017 (23 weeks after the release of 13RW’s first season)	- Setting: 41% outpatient, 41% inpatient, 17% emergency department. - Timing: 73% in the first 8 weeks after the release. - Diagnosis: 97% any mental health diagnosis, 84% major depressive disorder. - Presentation: 52% suicidal ideation, 19% suicide attempt. - In 21% and 32% of references (guardians and patients, respectively), 13RW was reported to have made a contribution to worsening mental health symptoms.
Romer, 2020 [44]	Number of suicides (global)	Same sample as in Bridge et al., 2020 [37]	Reanalysis of the results from Bridge et al., 2020 [37], performed by using a simple auto-regression model that removed auto-correlation and national trends in suicide	- Males: no increase in the first month (values were similar to the month before the release). - Females: small increase, though non-significant, in the first month after the release. - No significant increases in other months.
Romer, 2022 [45]	Number of suicides (youths)	Suicide data from individuals aged 10 to 29 in the period 2013–18 (data obtained from the CDC’s National Vital Statistics System), USA	Ecological study comparing suicide rates in the years of the release of 13RW’s first and second season with expected values	- No effect of 13RW in youth suicides once seasonality, auto-correlation, and longer-term trends were removed. - The increase in male adolescent suicides detected in other studies [37,41,46] was attributed to seasonality.
Sinyor et al., 2019 [46]	Number of suicides (youths)	264 suicides in people aged 29 and under, identified through the Office of the Chief Coroner of Ontario, Canada	Ecological study comparing suicide rates in the 9 months after the release of 13RW’s first season with expected values	- 40 excess suicides (+18%). - 24% and 23% increase among 10- to 19- and 20- to 29-year-olds, respectively, compared to the previous 3 years. - 24% and 23% increase in males and females, respectively, across both age groups. - No increase in the cutting method.
Sinyor et al., 2022 [47]	Emergency department admissions for self-harm (youths and young adults)	Emergency department visits for self-harm from patients aged 10 to 45 in the period 2007–18 (data extracted from the National Ambulatory Care Reporting System, the Ontario Health Insurance Plan Claims Database, and the Registered Persons Database), Ontario, Canada	Ecological study comparing emergency department visits for self-harm in the first year after the release of 13RW’s first season with expected values and with admissions from previous years	- Significant increase in self-harm-related emergency department admissions in the 3 months after the release, continuing for 5 months. - Compared with expected values: 75 excess visits (+6.4%). 60 excess visits in 10- to 19-year-olds. 85 excess visits in females. No excess visits in other groups. - Compared with values from the previous year: 10- to 19-year-olds: 290 excess visits (+25.4%); 20- to 29-year-olds: 169 excess visits (+17%); 30- to 45-year-olds: 174 excess visits (+21.2%).
Sugg et al., 2019 [48]	Use of Crisis Text Line (CTL), a technology-based platform for crisis events	584,157 total CTL conversations, USA	Ecological study comparing the prevalence of daily CTL use in the 36 days after the release of 13RW’s second season with expected values	- Excess of 4302 CTL conversations (first 18 days after the release) and 20,500 (first 36 days, 4 times higher). - Sub-analysis of suicidal thought-related conversations: excess of 4,782 (+31%) in the 36 days after. - The suicides of Anthony Bourdain and Kate Spade (18th day after the release) had a greater impact on CTL conversation volume than 13RW (2.7 times higher). - Differences between seasons 1–2 were attributed to the promotion of help-seeking behaviors before the release of season 2.
Thompson et al., 2019 [49]	Use of Crisis Text Line (CTL), a technology-based platform for crisis events	70,330 total CTL conversations, USA	Ecological study comparing the prevalence of daily CTL use after the release of 13RW’s first season with expected values	- Brief increase in CTL conversation volume (days 5–6 after the release), followed by a much more substantial and significant decrease (12.7% in the 18 days after, 29.5% in the 49 days after). - Sub-analysis of suicidal thought-related conversations: similar decline during the same period.

In the pediatric population, all the available studies agreed that hospital and emergency department admissions for suicidal reasons increased in the first month/s after the show’s release [38,39,40,47], particularly among adolescents [40,47]. Whereas it was controversial if the raise in emergency department visits occurred in females [47] or in both gender groups [40], excess inpatient hospitalizations were described only upon females, particularly among those who had been identified as ‘Black’ in terms of ethnicity [40]. In clinical contexts, most references to 13RW were registered in individuals with current mental health diseases (97%), with major depressive disorder being the most prevalent issue (84%) [43].

Two studies analyzed the volume of Crisis Text Line (CTL) conversation, a technology-based platform for crisis events, after the release of the first [49] and second [48] seasons of 13RW, respectively. Whilst season 1 was followed by a significant decrease in CTL conversation volume, season 2 experienced a marked increase in the same outcome. These differences were attributed to the promotion of help-seeking behaviors before the release of the second season [48].

Google searches for suicide also increased significantly [36] and, in terms of the social network Twitter, the USA and the UK were the countries with more 13RW publications—‘tweets’—discussing suicide [42].

Five studies regarding suicide mortality rates were detected. Three of them reported a significant increase of about 20% in adolescents (10- to 19- or 10- to 17-year-olds) in comparison with the expected values [37,41,46]. Specifically, Bridge et al. revealed a higher amount of deaths in the first month post-release (28.9%) [37]. Conversely, another author, Romer, disagreed with these results in two publications: in the first one, he re-analyzed the results of Bridge et al., concluding there was no statistically significant increase after removing trends and auto-correlation [44]; in the second, he went beyond and attributed the differences detected by the other authors to a seasonality effect [45]. No solid conclusion was reached in terms of gender: whereas Sinyor et al. and Niederkrotenthaler et al. reported a significantly higher impact in females [41,46], Bridge et al. described the same phenomenon in males [37]. Romer, however, detailed a positive but non-significant increase of incidence in girls during the month following the release in his first article [44] and no increase at all for either sex group in the second one [45]. As for method used, the hanging choice raised significantly, but the prevalence of cutting and firearm use was not altered [41,46]. Although one of the studies also outlined an excess of suicides in young adults (20- to 29-year-olds) [46], the rest did not report any extra suicide mortality in other age groups apart from among adolescents.

### 3.2. Outcomes in Association with 13RW Based on Subjective Measures (See Table 3)

Thirteen publications registered effects in relation to suicide, all of them by using self-reported methods on 13RW viewers. Most studies comprised adolescent samples [50,51,52,53,54,55,56,57,58], whilst others included young adults [50,51,59,60,61] and even parents or guardians [50,51,53,54,62]. Only two studies were based on clinical populations [54,55] as the rest of them were performed with vulnerable subjects recruited through social networks [52,58], survey firms [50,51,59,62] or other procedures [53,56,57,60,61].

Several authors described positive outcomes after watching the series. Some of the items that were noted after 13RW exposure were better understanding of depression/suicide, positive behavioral changes (in adolescents and young adults) [51], and increased suicide knowledge and suicide stigma reduction (in young adults) [60]. The series appeared to increase the likelihood to have conversations with parents about suicide and to seek information on the matter [51]; parents also reported better understanding of the topic, more comfort at discussing it, and a greater likelihood of prompting conversation [62]. The number of viewed episodes was positively correlated with perceived norms about mental illness that, in the end, were related to changes in pro-social mental health behaviors [50]. In contrast, the only study that used a randomized design linked the increase in the frequency of conversation and information seeking about suicide to presenting depressive symptoms after watching the show [57].

Audiences of 13RW were more likely to be female [55] and to have previously presented more self-harm behaviors [52,55]. About 50% of viewers expressed negative reactions to 13RW [55] or believed it had increased their risk of suicide [54]. Mood worsening after 13RW was associated with premorbid sadness and lack of motivation and also implied a higher suicide risk [52]. Being a student was associated with higher suicide risk but, at the same time, higher suicide acceptance [59].

The increase in suicidal ideation that has been associated with the show was not homogeneous across studies and varied depending on individual factors. Higher identification with 13RW’s main character [54,55], real-life suicide exposure [53], having a previous history of suicidal thoughts [58], watching only some of the episodes of the series [59], and watching 13RW during a suicide cluster [56] were positively correlated with suicidal ideation. However, two studies found no differences in suicidal ideation before and after watching the show [55,60], and another one even reported a protective effect [53]. In the case of self-reported suicide attempts committed after 13RW exposure, no significant association was reported.

One study developed an artificial intelligence tool which could be administered to university students in a quasi-experiment, showing no statistically significant differences in the associations of the self-concept ‘SUICIDE’ and exposure to 13RW [61].

**Table 3 ijerph-20-05270-t003:** Summary of the articles reporting outcomes in association with 13RW based on subjective measures.

Article	Dependent Variable(s)	Sample	Design	Results
Arendt et al., 2019 [59]	Suicide risk Optimism Suicide acceptance Self-harm Suicidal ideation Intention to help a suicidal person	729 individuals aged 18 to 29, who reported having access to Netflix. Recruited by the survey firm Qualtrics, USA	Survey administration 10 days before and 1 month after the release of 13RW’s second season	- Suicide risk: increased in current students and people who only watched some of the episodes. - Optimism: increased in those who watched the entire season; decreased in those who only watched some of it. - Suicide acceptance: increased in students; decreased in females. No relation with watching 13RW. - Self-harm and suicidal ideation: increased in students who only watched some of the episodes and non-students who watched some or all the episodes; decreased in students who watched all the episodes. - Intention to help a suicidal person: increased in non-students and those who watched all the episodes; decreased in those who only watched some of the episodes.
Carter et al., 2020 [50]	Perceived norms about mental illness	Same sample as in Lauricella et al., 2018 [51]	Further analysis of the results from Lauricella et al., 2018 [51], by using nested measurement invariance models	- The number of 13RW viewed episodes was positively correlated to perceived norms about mental illness. - In viewers, these norms were related to change in pro-social mental health behaviors (manifested distinctively based on age and regional factors).
Chesin et al., 2020 [60]	Suicidal ideation Suicidal behavior Suicide knowledge Suicide stigma Depressive symptoms	818 college students (522 of whom had watched 13RW) aged 18 to 25 and recruited online, USA	Survey administration several months after the release of 13RW’s first season, comparing those who had watched the series to those who had not	- Suicidal ideation and suicidal behavior: no differences. - Suicide knowledge: increased in 13RW viewers, particularly among those without personal exposure to suicide. - Suicide stigma: decreased in 13RW viewers. - Self-reported depressive symptoms: increased in 13RW viewers.
Cingel et al., 2021 [62]	Parent-sensitive topic understanding, comfort and conversation prompting about suicide and other mental health issues	778 parent viewers of 13RW, obtained from the same sample as in Lauricella et al., 2018 [51]	Further analysis of the results from Lauricella et al., 2018 [51]	- Positive relation between topic understanding and comfort discussing these topics. - Positive relation between comfort discussing these topics and prompting a conversation. - Stronger association among parents residing in Australia, the USA, and the UK, in comparison to parents in Brazil.
da Rosa et al., 2019 [52]	Suicide risk Mood changes	7004 individuals aged 12 to 18 who reported having watched 13RW’s first season, recruited through posts on 13RW-themed social media groups, Brazil	Survey administration 54–71 days after the release of 13RW’s first season	- Suicide risk (suicidal thoughts, self-harm or suicide attempts): only 25% reported never to have had any suicidal behavior or ideation. - Mood: 32.1% improved, 23.7% worsened; significantly associated with feelings of sadness and lack of motivation before 13RW and suicide risk after 13RW.
Ferguson, 2021 [53]	Suicidal ideation Depressive symptoms	174 individuals aged 11 to 18 and their primary caregivers, recruited online or by family members from the university, USA	Survey administration with no time relation to the release of 13RW (October 2017–May 2019), analyzed with regression models	- Exposure to 13RW in youths: reduced depression, reduced suicidal ideation (both significant). - Exposure to 13RW in parents: reduced depression (non-significant), no relation with suicidal ideation. Real-life suicide exposure was the only significant predictor of suicidal ideation. - Models including 13RW were better than the ones without this variable.
Hong et al., 2019 [54]	Perceived effects of watching 13RW	87 individuals aged 10 to 17 who presented to a psychiatric emergency department with a suicide-related concern from July 2017 to March 2018 (49% of whom had watched 13RW), and their parents (n=87), USA	Completion of a battery of self-reported measures during the emergency department visit	- 51% of viewers believed 13RW increased their suicide risk. - 39 viewers reported increased negative affect after 13RW and 40 referred increased positive affect. - Identification with 13RW’s main character significantly correlated with higher suicidal ideation and depressive symptoms. - Likelihood to talk to others about mental health after 13RW: 28% increased, 28% decreased. Those who reported an increase were more likely to believe 13RW had not raised their suicide risk.
Lauricella et al., 2018 [51]	Perceived effects of watching 13RW	1722 teens (aged 13 to 17), 1798 young adults (aged 18 to 22) and 1880 parents who completed an online survey, recruited by IPSOS Research; Australia/New Zealand, Brazil, UK, and USA	Survey administration several months after the release of 13RW’s first season, comparing those who had watched the series to those who had not	- Better understanding of depression and suicide after 13RW: 59–88% teens and young adults, 44–83% parents. - Positive behavioral changes after 13RW: 67–76% teens and young adults. - Information seeking after 13RW: 38–68% (depression) and 36–65% (suicide) in teens and young adults. - 13RW helped having conversations with parents about suicide and depression (56–71%). - Individual characteristics (age, resilience, social anxiety) influence responses to the show.
Nesi et al., 2020 [55]	Perceived effects of watching 13RW Suicidal ideation Suicide attempts Non-suicidal self-injury (NSSI)	242 individuals (122 of whom had watched 13RW) aged 11 to 18, hospitalized in a psychiatric inpatient facility between May 2018 and January 2019, USA	Completion of a battery of self-reported measures at hospital admission	- Females: more likely to have watched 13RW (63.3%). - 55.9% expressed negative reactions to the show, while 33.8% expressed positive reactions. - Having watched 13RW was associated with greater likelihood of past-year NSSI (2.53 times higher), but not with suicidal ideation or suicide attempts. - Greater identification with and perceived likeability of the main character were associated with suicidal ideation and past-year NSSI.
Swedo et al., 2021 [56]	Suicidal ideation and suicide attempts during the suicide cluster that was reported in Ohio from August 2017 to March 2018	15,083 students (7th- to 12th-grade) from 34 public schools (recruited by the local health department and Ohio Department of Health), USA	Survey administration in April-May 2018, 13–14 months after the release of 13RW’s first season and 1–2 months after the second	- Watching 13RW before the cluster: no significant association with suicidal ideation or suicide attempts. - Watching 13RW during the cluster: increased suicidal ideation (AOR 1.4), specifically higher for students without previous history of suicidal ideation (AOR 1.5). No association with suicide attempts.
Uhls et al., 2021 [57]	Social and mental health issues Suicidal ideation	157 individuals aged 13 to 17, recruited by a panel-based research platform operated by the National Opinion Research Center at the University of Chicago, USA	Survey administration before (T1) and one month after (T2) the release of 13RW’s third season, randomizing participants to intervention group (watching 13RW) and non-intervention group	- Increased likelihood to discuss suicide among viewers at T2 and from T1 to T2. - Higher levels of depressive symptoms at T2: more frequent conversations about suicide from T1 to T2. - Reported little interest or pleasure in doing things at T2: increased likelihood to search information about suicide.
Wang et al., 2022 [61]	Distance between the concept ‘SUICIDE’ and other topics in cognitive mapping	81 college freshmen from a public university, USA	Administration of a tool which had been created with artificial intelligence (by using the Galileo method) six months after the release of 13RW’s first season to compare those who had watched the series to those who had not	- No differences between viewers and non-viewers. - Female viewers perceived themselves to be closer to the concept ‘SUICIDE’ than male viewers.
Zimerman et al., 2018 [58]	Suicidal ideation Depressive symptoms	21,062 individuals, aged 12 to 19, who reported having watched all the episodes of 13RW’s first season, recruited through Facebook advertising, USA and Brazil	Survey administration before and after the release of 13RW’s first season	- Suicidal ideation in subjects with a previous history of suicidal thoughts: 16.5% increased, 59.2% decreased, 24.2% unchanged. - Suicidal ideation in subjects without a previous history of suicidal thoughts: 6.4% increased, 93.6% unchanged.

### 3.3. Risk of Bias and Quality Assessment (See Table 4 and Table 5)

In general, the quality of the included studies is moderate; still, some methodological difficulties have been detected in most publications. Three studies reported conflicts of interest, as they were developed by the same research group that received indirect funding on behalf of Netflix; these studies highlight the potential positive effects of the series [50,51,62]. In many articles, there is also a surprising lack of references to an ethical approval by institutional committees and an absence of proper informed consent beyond the passive acceptance of participants.

**Table 4 ijerph-20-05270-t004:** Risk of bias and quality assessment (AXIS tool results) of the articles reporting outcomes in association with 13RW based on objective measures.

		Ayers et al., 2017 [36]	Bridge et al., 2019 [37]	Cooper et al., 2018 [38]	Feuer and Havens, 2017 [39]	Le et al., 2022 [40]	Niederkrotenthaler et al., 2019 [41]	Pérez-Martínez et. al, 2020 [42]	Plager et al., 2019 [43]	Romer, 2020 [44]	Romer, 2022 [45]	Sinyor et al., 2019 [46]	Sinyor et al., 2022 [47]	Sugg et al., 2019 [48]	Thompson et al., 2019 [49]
** *Introduction* **															
1	Were the aims/objectives of the study clear?	Yes	Yes	Yes	No	Yes	Yes	Yes	Yes	Yes	Yes	Yes	Yes	Yes	Yes
** *Methods* **															
2	Was the study design appropriate for the stated aim(s)?	Yes	Yes	Yes	No	Yes	Yes	Yes	No	Yes	Yes	Yes	Yes	Yes	Yes
3	Was the sample size justified?	No	No	No	No	No	No	No	No	No	No	No	No	No	No
4	Was the target/reference population clearly defined? (Is it clear who the research was about?)	No	Yes	Yes	No	Yes	Yes	No	No	Yes	Yes	Yes	Yes	Yes	No
5	Was the sample frame taken from an appropriate population base so that it closely represented the target/reference population under investigation?	No	Yes	Yes	No	Yes	Yes	No	No	Yes	Yes	Yes	Yes	Yes	No
6	Was the selection process likely to select subjects/participants that were representative of the target/reference population under investigation?	Yes	Yes	Yes	Yes	Yes	Yes	No	No	Yes	Yes	Yes	Yes	Yes	No
7	Were measures undertaken to address and categorize non-responders?	No	No	No	No	No	No	No	No	No	No	No	No	No	No
8	Were the risk factor and outcome variables measured appropriate to the aims of the study?	Yes	Yes	Yes	No	Yes	Yes	Yes	Yes	Yes	Yes	Yes	Yes	Yes	Yes
9	Were the risk factor and outcome variables measured correctly using instruments/measurements that had been trialed, piloted or published previously?	No	Yes	No	No	Yes	No	No	No	Yes	Yes	Yes	Yes	Yes	No
10	Is it clear what was used to determined statistical significance and/or precision estimates? (e.g., p values, CIs)	Yes	Yes	Yes	No	Yes	Yes	No	No	Yes	Yes	Yes	Yes	Yes	Yes
11	Were the methods (including statistical methods) sufficiently described to enable them to be repeated?	Yes	Yes	Yes	No	Yes	Yes	No	No	Yes	Yes	Yes	Yes	Yes	Yes
** *Results* **															
12	Were the basic data adequately described?	Yes	Yes	Yes	No	Yes	Yes	Yes	Yes	Yes	Yes	Yes	Yes	Yes	Yes
13	Does the response rate raise concerns about non-response bias?	Yes	Yes	Yes	Yes	Yes	Yes	Yes	Yes	Yes	Yes	Yes	Yes	Yes	Yes
14	If appropriate, was information about non-responders described?	No	No	No	No	No	No	No	No	No	No	No	No	No	No
15	Were the results internally consistent?	Yes	Yes	Yes	Yes	Yes	Yes	Yes	Yes	Yes	Yes	Yes	Yes	Yes	Yes
16	Were the results for the analyses described in the methods, presented?	Yes	Yes	Yes	Yes	Yes	Yes	Yes	Yes	Yes	Yes	Yes	Yes	Yes	Yes
** *Discussion* **															
17	Were the authors’ discussions and conclusions justified by the results?	Yes	Yes	Yes	Yes	Yes	Yes	Yes	Yes	Yes	Yes	Yes	Yes	Yes	Yes
18	Were the limitations of the study discussed?	No	Yes	Yes	No	Yes	Yes	No	No	Yes	Yes	Yes	Yes	Yes	Yes
** *Other* **															
19	Were there any funding sources or conflicts of interest that may affect the authors’ interpretation of the results?	No	No	No	No	No	No	No	No	No	No	No	No	No	No
20	Was ethical approval or consent of participants attained?	No/No	No/No	Yes/No	No/No	No/Yes	No/No	No/No	Yes/Yes	No/No	No/No	Yes/No	No/No	Yes/No	No/No

**Table 5 ijerph-20-05270-t005:** Risk of bias and quality assessment (AXIS tool results) of the articles reporting outcomes in association with 13RW based on subjective measures.

		Arendt et al., 2019 [59]	Carter et al., 2020 [50]	Chesin et al., 2020 [60]	Cingel et al., 2021 [62]	da Rosa et al., 2019 [52]	Ferguson, 2021 [53]	Hong et al., 2019 [54]	Lauricella et al., 2018 [51]	Nesi et al., 2020 [55]	Swedo et al., 2021 [56]	Uhls et al., 2021 [57]	Wang et al., 2022 [61]	Zimerman et al., 2018 [58]
** *Introduction* **														
1	Were the aims/objectives of the study clear?	Yes	Yes	Yes	Yes	Yes	Yes	Yes	Yes	Yes	Yes	Yes	No	Yes
** *Methods* **														
2	Was the study design appropriate for the stated aim(s)?	Yes	Yes	Yes	Yes	Yes	Yes	Yes	Yes	Yes	Yes	Yes	No	Yes
3	Was the sample size justified?	No	Yes	No	No	No	No	No	No	No	No	No	No	No
4	Was the target/reference population clearly defined? (Is it clear who the research was about?)	Yes	Yes	Yes	Yes	Yes	Yes	Yes	Yes	Yes	Yes	Yes	No	Yes
5	Was the sample frame taken from an appropriate population base so that it closely represented the target/reference population under investigation?	Yes	Yes	Yes	Yes	Yes	Yes	Yes	Yes	Yes	Yes	Yes	Yes	Yes
6	Was the selection process likely to select subjects/participants that were representative of the target/reference population under investigation?	Yes	Yes	Yes	Yes	Yes	Yes	Yes	Yes	Yes	Yes	Yes	Yes	Yes
7	Were measures undertaken to address and categorize non-responders?	Yes	No	No	No	No	No	No	No	No	No	Yes	No	No
8	Were the risk factor and outcome variables measured appropriate to the aims of the study?	Yes	Yes	Yes	Yes	Yes	Yes	Yes	Yes	Yes	Yes	Yes	Yes	Yes
9	Were the risk factor and outcome variables measured correctly using instruments/measurements that had been trialed, piloted or published previously?	Yes	No	Yes	No	No	Yes	Yes	Yes	Yes	Yes	Yes	No	Yes
10	Is it clear which methods were used to determined statistical significance and/or precision estimates? (e.g., *p* values, CIs)	Yes	Yes	Yes	Yes	Yes	Yes	Yes	Yes	Yes	Yes	Yes	No	Yes
11	Were the methods (including statistical methods) sufficiently described to enable them to be repeated?	Yes	Yes	Yes	Yes	Yes	Yes	Yes	Yes	Yes	Yes	Yes	No	Yes
** *Results* **														
12	Were the basic data adequately described?	Yes	Yes	Yes	Yes	Yes	Yes	Yes	Yes	Yes	Yes	Yes	Yes	Yes
13	Does the response rate raise concerns about non-response bias?	No	Yes	Yes	Yes	Yes	Yes	Yes	Yes	Yes	Yes	No	Yes	Yes
14	If appropriate, was information about non-responders described?	Yes	No	No	No	No	No	No	No	No	No	Yes	No	No
15	Were the results internally consistent?	Yes	Yes	Yes	Yes	Yes	Yes	Yes	Yes	Yes	Yes	Yes	Yes	Yes
16	Were the results presented for the analyses described in the methods?	Yes	Yes	Yes	Yes	Yes	Yes	Yes	Yes	Yes	Yes	Yes	Yes	Yes
** *Discussion* **														
17	Were the authors’ discussions and conclusions justified by the results?	Yes	Yes	Yes	Yes	Yes	Yes	Yes	Yes	Yes	Yes	Yes	No	Yes
18	Were the limitations of the study discussed?	Yes	Yes	Yes	Yes	Yes	Yes	Yes	No	Yes	Yes	Yes	No	Yes
** *Other* **														
19	Were there any funding sources or conflicts of interest that may have affected the authors’ interpretation of the results?	No	Yes	No	Yes	No	No	No	Yes	No	No	No	No	No
20	Were ethical approval or consent of participants attained?	Yes/No	Yes/Yes	Yes/No	Yes/Yes	Yes/Yes	Yes/Yes	Yes/Yes	No/No	Yes/No	No/Yes	Yes/Yes	Yes/No	Yes/No

## 4. Discussion

The aim of the current study was to provide a review of the evidence on the role of 13RW as an influencing factor for suicide. Whereas previous analyses of other media productions have mainly focused on the suicide mortality rate, other variables—such as suicidal ideation or actual suicide attempts—have traditionally been neglected or, at least, not analyzed deeply enough. In the case of 13RW, the amount of studies focusing on several outcomes (not only deaths by suicide) has made it possible to analyze the problem from several angles, which can be considered a strength. Few dramatized depictions of suicide have been object of such interest and controversy, and in our era of information, the quantity of available publications has been considerable compared to past experiences.

Still, several limitations need to be taken into account. As mentioned above, a publication bias cannot be excluded when considering the media dimension of the series. This fact can be inferred from the proportionally low number of original research articles that have been published, in comparison with the amount of publications that have reviewed the issue and cited the few available studies over and over again. Nearly all the articles were performed in the USA and only a small proportion considered samples from other countries (Canada [46,47], Brazil [50,51,52,58,62], Australia, New Zealand, or the UK [50,51,62]), which can affect the generalization of the results. The number of individuals included for each study was generally low and mainly comprised adolescents and young adults, with small regard given to other age groups. Most authors used retrospective observational designs, implying methodological limitations that make it difficult to detect a causal relationship between the series and suicidality; additionally, some studies suggest that the observed effect of 13RW may not be specific to the series but attributable to exposure to suicidal images or news in general. Most studies are hardly replicable, being based on information that was obtained in a certain historic moment and which cannot be understood out of this context; in other words, it is impossible to isolate the crude effect of the series from the rest of social confounding factors that are known to have an influence on suicide rates: financial crisis, poverty, social changes in adolescence, migration, drug abuse, political challenges... and, consequently, the interpretation of the results requires extreme caution. Regarding the methodology of our study, the cross-sectional quality assessment tool that was used was not entirely suited to evaluate all of the studies included, as not all of them had a cross-sectional design [57,61].

In general terms, objective sources support the presence of a Werther effect following the release of 13RW. Regarding suicide mortality, most authors report an increase in the number of suicides completed by adolescents after the series’ release [37,41,46], and one of them also describes an increase in deaths by suicide in the group of young adults [46]. This higher repercussion among teenagers is in line with the Werther effect hypothesis, especially considering Hannah Baker’s age and the audiences for whom the show was intended, and it might have been aggravated if we consider that adolescence is a period of increased vulnerability, as individuals of such ages lack the emotional and cognitive strategies to cope with certain experiences or stressing factors. In addition, adolescents tend to search for behavioral reference models to identify with, which are generally outside the family environment and are often found in the media.

On the contrary, two other studies written by the same author report no significant increase in suicide mortality after the series premiered [44,45]. Nevertheless, when Romer’s first article [44] is examined in detail, it is found that it is not an original research work, as it re-analyses Bridge et al.’s results [37] by using a different statistical approach. A letter exchange between both authors is available, with each trying to justify why their methodology is more precise and criticizing the other’s conclusions [63,64]. One of Romer’s criticisms of Bridge at al.’s work is based on the fact that the increase in excess suicides seems to have started one month before 13RW was released. Bridge attributed such a trend to different factors, such as (1) the situation of show promotion, suggesting that promotional material may contribute to suicide dissemination as well as the fiction itself; and (2) the fact that, although the series premiered the 31 March 2017, suicide debate had already started by then, creating social concern about the problem and turning it into a ‘hot topic’ with contagious potential. In a second study published more recently with a new source of data [45], Romer used a statistical method that controlled for seasonal changes and auto-correlation to conclude that 13RW had not increased suicide rates, again opposing the results of other authors [37,41,46]; nevertheless, no formal response has been published yet.

Contrary to expectations, no clear pattern has been identified in terms of gender, with some authors describing a greater impact of suicide mortality in females [41,44], males [37], both [46], or no groups [45]. As previous studies of the impact of fictionalized suicides reported an increased vulnerability in subjects with the same age and gender as the victim [7,8,9,10,11], it needs to be discussed why, in the case of 13RW, most studies give credit to the first statement but not to the second. The first possible explanation is a circumstance that has been repeatedly reported in literature: whereas females tend to attempt suicide more often, males are more successful at achieving it [65]. A second supposition is the fact that, surprisingly, Hannah’s suicide is not the only self-harm gesture that is pictured in the series: during the last seconds of the final episode of 13RW’s first season, Alex, a male classmate of Hannah’s, attempts suicide by gunshot—even though his action is dramatized in a more discrete (and undoubtedly less relevant) way than hers. A psychobiological hypothesis should also be taken into account if we consider that 13RW mainly affected adolescents, a critical age range where gender identity is still under development [66]. As contemporary youth populations are more diverse and globalized, their identification patterns go beyond the classic conventions of previous social models [67,68]. This brings into question whether the concept of identification with the main character transcends age and gender, or is better defined by other characteristics with which vulnerable individuals can empathize, such as the experiences of bullying, aggression and relationship problems that Hannah Baker suffered. Further research is required to identify which characteristics of fictional suicide portrayals may lead to contagion and to what extent they play a role in this phenomenon. On the other hand, it is also remarkable that wrist-cutting, the method chosen in the series, did not experience a raise among subjects who attempted suicide following exposure to the show, which would also have been in line with a Werther effect.

According to published evidence, 13RW not only might have increased suicide mortality in adolescents, but also the need of medical assistance for psychiatric reasons in this population [38,39,40,43,47], suggesting a global repercussion of the series in the suicidality spectrum, from suicidal ideation to actual attempts. This effect has also been described in younger population (6- to 9-year-olds with “Other” races/ethnicities), but not in age groups over 19. As it occurred with suicide mortality, gender predominance obtained mixed results, with most reports describing increased rates of medical assistance in females.

Self-reported responses to the show, mostly assessed by the articles that used subjective measures, obtained much more heterogeneous results: whilst some authors describe an increase in suicidal symptoms and behaviors, others report no association at all or even an anti-suicidal effect. Nevertheless, most of them agree that individual features affect viewers differently, modulating their reactions. Many of these risk factors had been described before (previous mental health history, higher identification with the suicidal character, etc.), but others were less known (exposure during a suicide cluster or watching only some episodes of the show).

Suggesting a Papageno effect, some publications report pro-mental health outcomes (such as stigma reduction or better understanding of depression and suicide), but some of these effects seem quite unspecific and hardly attributable to the series. On the other hand, the methodological limitations of the studies based on subjective measures are considerable and their conclusions, though interesting and suggestive, must be dealt with carefully. Collecting information by indirect methods (surveys and other self-report measures) implies a memory bias, as the answers are inherently influenced by the person’s recall of the show and not only by the show itself. The fact that 13RW received generally favorable reviews from audiences and critics, consequently, may have caused the underestimation of its negative effects and the overestimation of the detection of positive reactions. For the future, focusing on prospective designs would be a better means of assessing the impact of positive and negative consequences instead of searching a global effect.

Other outcomes that have emerged from the articles included in this review are the increase in Google searches for suicide [49] and the increase in Twitter posts discussing this topic [42] that followed the show’s release. Such evidence, combined with the results of the rest of studies, suggests that 13RW probably had enough potential to trigger a social interest in the subject, one that was amplified and disseminated through media like a wave. This might have induced an anti-suicidal effect in general population, but also a pro-suicidal effect in vulnerable subjects. Consequently, these pro-suicidal and anti-suicidal effects of 13RW may not be attributed to the fiction alone, but also to the social context that was created because of the series, where media had a clear influence. An example of this phenomenon is the variation of CTL volume conversation across 13RW season 1 and season 2 [48,49], which could be used as a clue to evaluate the impact of the series with and without a suicide prevention campaign—a strategy that was used only for the second season. Although the series and the social context were similar in both cases, the approach from the media differed and such variations might have altered the audiences’ responses. At this point, it may be sensible to wonder if the approach to the topic in the second case might have contributed to spreading help-seeking behaviors and to what extent they reduced the impact on suicidality. 

When products which are aimed at sensitive audiences are being prepared, it is highly advisable to seek specialized counseling and to prevent vulnerable subjects from suffering possible adverse effects on their emotional stability and related behaviors. Netflix itself decided to apply these changes after detecting the first indices of negative outcomes presumably attributed to the show. From our point of view, it would not be difficult to integrate educational consultants into creative teams to confirm that media material fulfills prevention standards before it is released. Interactive warnings before each episode could be used to ask viewers if they are—or have been—under specialized follow-up for suicidal reasons, and to remit them to their therapeutic referents for advice in case they feel the show might worsen their mental stability.

It has also been proposed that 13RW could be used as supporting material for suicide prevention campaigns and even as an educational tool in high schools and medical centers [69,70,71,72]. However, evidence suggests that its potential employment is clearly undermined by the risk of affecting vulnerable subjects and, certainly, we would not recommend its use outside of an educative setting that contextualizes the show and provides proper information about suicide and its portrayal in the series (see Table 1).

The discussion of the current study would be incomplete without mentioning a previous systematic review, published in April 2022, that focused on the association between 13RW’s first season and suicidal ideation/behaviors, mental health symptoms, and help-seeking behaviors in youths [73]. The study was instructive and well-written, used a robust methodology and reached conclusions that were in line with ours in general terms. The authors included references up to January 2022 and, from an initial search of 645 studies, they found 17 publications that fulfilled inclusion criteria. In contrast, the present review included articles up to January 2023, considered all age groups and exposure to any season of the series, excluded qualitative designs, and used a different search strategy. That led to 496 studies, 27 of which could be accepted for data extraction: as a result, 12 non-overlapping references could be added to the synthesis [39,40,42,44,45,48,51,53,57,59,61,62]. Although it could be argued that a previous systematic review undermines the potential originality of the current study, the subject is so complex and heterogeneous that an updated review, in the authors’ judgement, enrichens and completes the previous results and discussion, contributes to expand the debate on the topic, and opens the door to further research.

One final consideration should be made. Although most research about 13RW has focused on how it may play a role in suicide dissemination, several authors have discussed the same phenomenon in the case of sexism [74,75], racism [75], cyberbullying [76], and sexual assault [77,78]. Such experiences have pointed out that, as for self-harm, aggressive behaviors also have a potential for imitation that might be influenced by media. This hypothesis raises questions on how collective responsibility, when combined with individual vulnerability, might contribute to the perpetuation—and even potentiation—of certain stereotypes that, in the end, result in discrimination and violence. Such issues deserve further investigation and should be addressed with properly designed studies.

## 5. Conclusions

The outcomes associated with 13RW have been mainly reported in individuals with the same age as the show’s protagonist (adolescents), showing scarce impact in other age groups. Most studies that analyzed data from objective sources suggest a Werther effect of the series (higher suicide mortality, higher medical admissions for suicidal reasons) and a higher interest for suicide in the network after the release. In contrast, studies based on subjective reports describe higher suicidal ideation in vulnerable individuals but also a possible Papageno effect in general population. However, this theoretical anti-suicidal effect would be offset by the tangible pro-suicidal consequences that have been attributed to the show. Methodological limitations are not to be disdained and, therefore, no causal association can be established between the series and suicidality. 

From a public health perspective, 13RW represents a fine example of suicide representation through media and its social and clinical repercussions. Primary prevention is the best strategy to address this problem and it should be particularly aimed at individuals with risk factors. Concrete measures to address the problem would be the revision of sensible material by qualified psychopedagogical personnel with a preventive, gender-based perspective; the promotion of psychoeducational programs among vulnerable viewers and their familiar, social and educational communities; and the use of warnings that alert of the risk before the show is displayed. Future fictional dramas planning to depict suicide should review this case to learn from its strengths and weaknesses, which will assist in contributing to the suicide prevention task.

## Figures and Tables

**Figure 1 ijerph-20-05270-f001:**
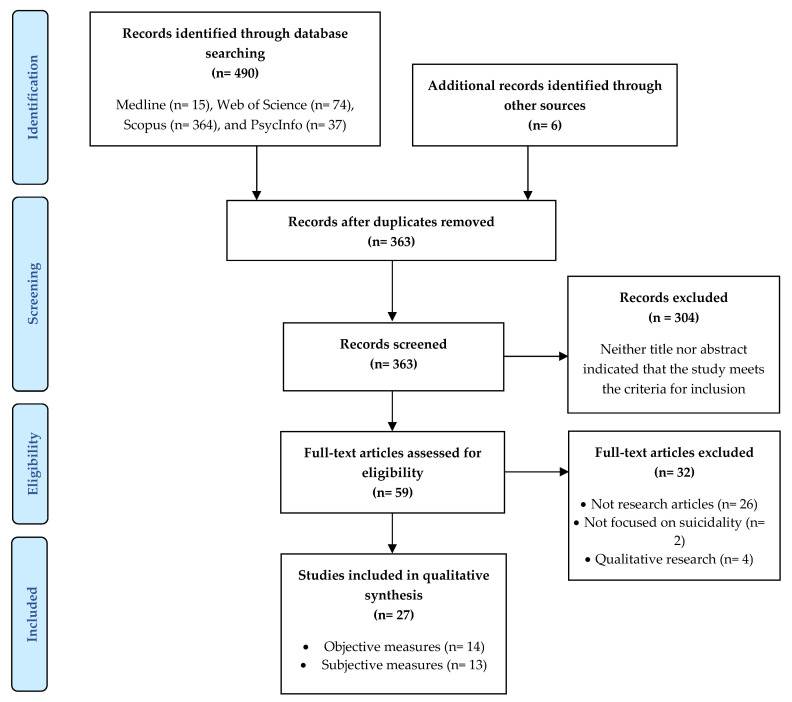
PRISMA flow chart with included and excluded studies, with reasons.

**Table 1 ijerph-20-05270-t001:** Myths and realities of suicide representation in 13RW.

Suicide in 13RW	Suicide in Clinical Practice
Social phenomenon	Health issue
Sense of success (personal, social)	Sense of failure (screening, case management)
External responsibility	Individual responsibility
Suicide to achieve control (over oneself and others)	Suicide as the result of a loss of control
Rational act	Irrational, illogical act
Inevitable	Evitable
Voluntary, adaptive and justifiable choice	Maladaptive, pathological reaction
No accompanying symptoms	Underlying mental disorder
Health system reinforces the problem	Health system as a potential solver of the problem
Suicide is the only alternative	Suicide is a symptom, not an alternative

## Data Availability

The data presented in this review are available upon request from the corresponding author.

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
