# Peer review of "Towards the Influence of Media on Suicidality: A Systematic Review of Netflix’s ‘Thirteen Reasons Why’"

_ijerph, 2023, doi:10.3390/ijerph20075270_

Round 1

Reviewer 1 Report

This paper is a systematic review of the effect of original studies analyzing the role of the online streaming series ‘Thirteen reasons why’ (13RW) as an influencing factor for suicide. Both positive and negative effects of watching 13RW were found. Although there had been a reported increase in help-seeking for mental health concerns, there were significant increases in suicidality and self-harm behaviors in vulnerable viewers. However, a causal relationship could not be established in the current study. Risk awareness is required for vulnerable viewers – it was suggested that psychoeducational programs should be created and delivered to fill this gap.

Comment 1: “In the case of fictional suicides, some authors have pointed out that vulnerable subjects, when exposed to films that depict suicide, experience a rise in suicidality that correlates with the level of identification with the main character”. Parasocial relationships have been described as coming from television and film, although there is more recent literature on this phenomenon in terms of social networking sites. It may help to refer to these relationships.

Comment 2: “Future fiction dramas planning to depict suicide should review this case to learn from its strengths and weaknesses, contributing to the suicide prevention task”. It would help to briefly list these strengths and weaknesses at the very end as well as in the highlights section. The main suggestion is psychoeducational programs. Perhaps elaborate more on what this may look like. What type of warnings can be suggested to capture the vulnerable persons’ attention?

Author Response

Sabadell, 20th of February 2023

Dear reviewer,

Thank you very much for your positive feedback.

Regarding your comments, hereby you will find our reply with the revisions you suggest.

  1. “In the case of fictional suicides, some authors have pointed out that vulnerable subjects, when exposed to films that depict suicide, experience a rise in suicidality that correlates with the level of identification with the main character”. Parasocial relationships have been described as coming from television and film, although there is more recent literature on this phenomenon in terms of social networking sites. It may help to refer to these relationships.

The phenomenon of parasocial relationships, closely linked to the concept of self-identification with a public character (fictional or not), is something that certainly must be mentioned in a debate like ours, as it can imply a higher self-harm risk when this character attempts suicide. As 13RW had a greater influence upon adolescents, and the use of social networking sites is more extended among them, their use may have implied an increased vulnerability. Accordingly, the sentence ‘In the era of social networking sites, parasocial relationships (and their influence on suicide models) need to be taken into consideration, especially in younger populations’ has been added to the Introduction. Three references on the phenomenon have been provided:

(1) Jarvi SM, Swenson LP, Batejan KL. Motivation for and use of social networking sites: Comparisons among college  students with and without histories of non-suicidal self-injury. J Am Coll Health. 2017;65(5):306-312. doi:10.1080/07448481.2017.1312410

(2) Hoffner CA, Cohen EL. Mental Health-Related Outcomes of Robin Williams’ Death: The Role of Parasocial  Relations and Media Exposure in Stigma, Help-Seeking, and Outreach. Health Commun. 2018;33(12):1573-1582. doi:10.1080/10410236.2017.1384348

(3) DeGroot JM, Leith AP. R.I.P. Kutner: Parasocial Grief Following the Death of a Television Character. Omega. 2018;77(3):199-216. doi:10.1177/0030222815600450

  1. “Future fiction dramas planning to depict suicide should review this case to learn from its strengths and weaknesses, contributing to the suicide prevention task”. It would help to briefly list these strengths and weaknesses at the very end as well as in the highlights section. The main suggestion is psychoeducational programs. Perhaps elaborate more on what this may look like. What type of warnings can be suggested to capture the vulnerable persons’ attention?

The strengths (anti-suicidal effect in general population) and weaknesses (pro-suicidal effect in vulnerable viewers) are already mentioned in the previous paragraph: ‘Most studies that analyzed data from objective sources suggest a Werther effect of the series (higher suicide mortality, higher medical admissions for suicidal reasons) and a higher interest for suicide in the network after the release. In contrast, studies based on subjective reports describe higher suicidal ideation in vulnerable individuals but also a possible Papageno effect in general population; however, this theoretical anti-suicidal effect would be offset by the tangible pro-suicidal consequences that have been attributed to the show. Methodological limitations are not to be disdained and, therefore, no causal association can be established between the series and suicidality.’

In the same direction, the highlights section also includes references to these strengths/weaknesses: ‘Positive and negative outcomes regarding suicide have been associated to the show’ and ‘Individual factors define vulnerable viewers that should be warned of the risk’. The authors preferred not to be more specific on these outcomes in order to avoid overlapping with Discussion and Conclusions.

Thank you for your suggestion of elaborating more on what specific measures could be carried out to address the problem, and the indication that they should be mentioned in the Conclusions. We added the sentence ‘Concrete measures to address the problem would be the revision of sensible material by qualified psychopedagogical personnel with a preventive, gender-based perspective; the promotion of psychoeducational programs among vulnerable viewers and their familiar, social and educational communities; and the use of warnings that alert of the risk before the show is displayed.’

Regarding these warnings, and your concerns on what type of warnings could be suggested to capture the vulnerable persons’ attention, we added the following sentence in the Discussion section: ‘Interactive warnings before each episode could be used to ask viewers if they are –or have been– under specialized follow-up for suicidal reasons, and to remit them to their therapeutic referents for advice in case they feel the show might worsen their mental stability’.

We hope the changes performed upon the manuscript fulfill the requirements of the journal; still, any extra contribution by the reviewers aiming to improve the quality of the paper is more than welcome. In case you need further clarification in any aspect, please do not hesitate to ask.

Thank you again for your interest and guidance.

We are looking forward to hearing from you.

Yours sincerely,

The authors

Reviewer 2 Report

I must commend the authors for such a great effort in producing the manuscript as well as the nice write-up. However, several sections of the manuscript need revision to enhance the quality of the manuscript.

INTRODUCTION

1.      The introduction seems ok except the aims. It would be better if there is a single aim but several objectives. Also, the “aims” provided were too generic/obvious for any review especially the first and third aims. I will strongly encourage the authors to set specific/measurable objectives that directly relates to the analysis, and the discussion.

METHOD

2.      Was the review registered?

3.      Exclusion criteria “be written in other languages”… authors should add “apart English, Spanish or Catalan”.

4.      Search strategy: If the movie was released on March 2017 why should the search starts from inception of the journal? Unless the authors may be comparing articles before March 2017 to after which may provide a comparative analysis and better argument.

5.      In case of disparity, agreement between authors was reached. Specifically how? 

RESULTS

6.      Presentation in Tables 2 and 3 should be in Alphabetical order.

7.      Usually the Tables are compilations that reflect the objectives. Therefore, the current Tables and the titles makes the manuscript more confusing.

8.      The Appraisal tool for Cross-Sectional Studies (AXIS tool) should be looked at again as authors reported some studies as being randomized control or quasi experiment in design

DISCUSSION.

The firs three paragraphs were like extension to introduction. Authors may simply cite the aims/objectives of the study at the beginning of the discussion and then summary of findings, then discuss it.

Author Response

Sabadell, 20th of February 2023

Dear reviewer,

Thank you very much for your positive feedback.

Regarding your comments, hereby you will find our reply with the revisions you suggest.

  1. The introduction seems ok except the aims. It would be better if there is a single aim but several objectives. Also, the “aims” provided were too generic/obvious for any review especially the first and third aims. I will strongly encourage the authors to set specific/measurable objectives that directly relates to the analysis, and the discussion.

As the main Aim of the study is already stated in the Abstract, the ‘aims’ in the Methods section have been changed to the ‘objectives’. The first objective has been extended by explaining what sources are going to be used, so that it fits better with Tables 2-3. After having considered your comments, the third aim has been deleted as it was too generic. We added another third objective (‘To discriminate whether individuals with the same features as 13RW’s protagonist (adolescent females) have been influenced differently by the possible effects of the show when compared to other subjects’) that we think is more specific and directly related to the topics commented in the Discussion.

  1. Was the review registered?

The protocol of the current review was submitted in Prospero in 2021 but, surprisingly, it was not approved by reviewers and we were not provided with the chance to modify it. We chose not to specifically state it in Methods; however, it can be added to the Discussion if you consider it is relevant.

  1. Exclusion criteria “be written in other languages”, authors should add “apart English, Spanish or Catalan”.

The sentence has been modified according to your suggestion.

  1. Search strategy: If the movie was released on March 2017 why should the search starts from inception of the journal? Unless the authors may be comparing articles before March 2017 to after which may provide a comparative analysis and better argument.

Although 13RW was released in March 2017, show promotion had started months (and even years) before. One of the studies that reached the final synthesis (Bridge et al., 2020) reported an increase of deaths by suicide in the previous month before the release, suggesting that promotional material has potential for suicide contagion as well as the show itself. Taking this information into account, we could not limit the pro-suicidal or anti-suicidal effect of the series only to the period that followed the release, so we chose not to set any restriction regarding publication year.

  1. In case of disparity, agreement between authors was reached. Specifically how?

For further clarification, the sentence has been changed as follows: ‘In case of disparity, each item of the scale was discussed by both reviewers until an agreement was reached.’

  1. Presentation in Tables 2 and 3 should be in Alphabetical order.

As you suggest, presentation in Tables 2 and 3, but also in Tables 4 and 5, has been changed to alphabetical order.

  1. Usually the Tables are compilations that reflect the objectives. Therefore, the current Tables and the titles makes the manuscript more confusing.

As reported above, the objectives of the study have been changed to fit better with the Tables.

To avoid confusion, the words ‘objective’ and ‘subjective’ (referred to the methods used to obtain the data) have been bolded and underlined in the title of each Table.

  1. The Appraisal tool for Cross-Sectional Studies (AXIS tool) should be looked at again as authors reported some studies as being randomized control or quasi experiment in design.

You are right. From the 27 final studies that contributed to data synthesis, only one of them was randomized and controlled (Uhls et al., 2021) and another used a quasi-experimental design (Wang et al., 2022). The rest of studies met criteria to be evaluated with the AXIS tool: to avoid multiplicities, the authors preferred to rate all the articles with the same method rather than finding different scales for each of these two studies. Still, it has to be admitted that using the tool upon these two articles can imply a limitation of the study and, if you consider this fact is relevant enough, it could be mentioned in the Discussion.

  1. The first three paragraphs of the discussion were like extension to introduction. Authors may simply cite the aims/objectives of the study at the beginning of the discussion and then summary of findings, then discuss it.

You are right. The first paragraph of the Discussion has been moved to the end of the Introduction, where we think it fits better.

As you recommended, we started the Discussion by citing the main aim of the study. The second and third paragraph have been re-formulated and amplified to discuss the strengths and limitations of the current work.

We hope the changes performed upon the manuscript fulfill the requirements of the journal; still, any extra contribution by the reviewers aiming to improve the quality of the paper is more than welcome. In case you need further clarification in any aspect, please do not hesitate to ask.

Thank you again for your interest and guidance.

We are looking forward to hearing from you.

Yours sincerely,

The authors

Reviewer 3 Report

This reminds me of a very similar discussion regarding suicides in fictive characters in television series in the late 1980:s. Like in this particular ms there were reports indicating increased suicides incidence after suicide in one of the main characters but Ron Kessler et al (1988 and 1989) showed that this increase was spurious. Interestingly they also explored the possibility that suicide events might even result in decreased suicides. I take the point raised by the authors of the present article that a streamed product which allows the persons to return to dramatic events may enforce the effect but this discussion in the late 80:s should be referred to.

Apart from that I feel this work is very well executed and the article is very well written

Author Response

Sabadell, 20th of February 2023

Dear reviewer,

Thank you very much for your positive feedback.

Regarding your comments, hereby you will find our reply with the revisions you suggest.

  1. This reminds me of a very similar discussion regarding suicides in fictive characters in television series in the late 1980:s. Like in this particular ms there were reports indicating increased suicides incidence after suicide in one of the main characters but Ron Kessler et al (1988 and 1989) showed that this increase was spurious. Interestingly they also explored the possibility that suicide events might even result in decreased suicides. I take the point raised by the authors of the present article that a streamed product which allows the persons to return to dramatic events may enforce the effect but this discussion in the late 80:s should be referred to.

Thank you for the references to Kessler’s work. As this contribution to the topic of suicide contagion through media is clearly relevant enough, the following sentence and references have been added to the Introduction: ‘Debate on the matter was already ongoing in the late 1980s, with some studies finding no reliable association between television coverage of suicides and self-inflicted deaths in adults (1) and teenagers (2).’

(1) Kessler R, Downey G, Stipp H, Milavsky J. Network Television News Stories about Suicide and Short-Term Changes in Total U.S. Suicides. J Nerv Ment Dis. 1989;177:551-555. doi:10.1097/00005053-198909000-00006

(2) Kessler R, Downey G, Milavsky J, Stipp H. Clustering of teenage suicides after television news stories about suicides: A reconsideration. Am J Psychiatry. 1988;145:1379-1383. doi:10.1176/ajp.145.11.1379

We hope the changes performed upon the manuscript fulfill the requirements of the journal; still, any extra contribution by the reviewers aiming to improve the quality of the paper is more than welcome. In case you need further clarification in any aspect, please do not hesitate to ask.

Thank you again for your interest and guidance.

We are looking forward to hearing for you.

Yours sincerely,

The authors
